# Microwave-Assisted Synthesis of N/TiO_2_ Nanoparticles for Photocatalysis under Different Irradiation Spectra

**DOI:** 10.3390/nano12091473

**Published:** 2022-04-26

**Authors:** Camilo Sanchez Tobon, Davor Ljubas, Vilko Mandić, Ivana Panžić, Gordana Matijašić, Lidija Ćurković

**Affiliations:** 1Faculty of Mechanical Engineering and Naval Architecture, University of Zagreb, 10000 Zagreb, Croatia; 2Faculty of Chemical Engineering and Technology, University of Zagreb, 10000 Zagreb, Croatia; vmandic@fkit.hr (V.M.); ipanzic@fkit.hr (I.P.); gmatijas@fkit.hr (G.M.)

**Keywords:** microwave-assisted synthesis, N/TiO_2_, ciprofloxacin, UVA, visible light, simulated solar light

## Abstract

Nitrogen-doped TiO_2_ (N/TiO_2_) photocatalyst nanoparticles were derived by the environmentally friendly and cost-effective microwave-assisted synthesis method. The samples were prepared at different reaction parameters (temperature and time) and precursor ratio (amount of nitrogen source; urea). The obtained materials were characterized by X-ray diffraction (XRD), photoelectron spectroscopy (XPS), Raman spectroscopy (RS), infrared spectroscopy (FTIR), diffuse reflectance spectroscopy (DRS), electron microscopy (SEM-EDS), and nitrogen adsorption/desorption isotherms. Two cycles of optimizations were conducted to determine the best reaction temperature and time, as well as N content. The phase composition for all N/TiO_2_ nanomaterials was identified as photoactive anatase. The reaction temperature was found to be the most relevant parameter for the course of the structural evolution of the samples. The nitrogen content was the least relevant for the development of the particle morphology, but it was important for photocatalytic performance. The photocatalytic activity of N/TiO_2_ nanoparticle aqueous suspensions was evaluated by the degradation of antibiotic ciprofloxacin (CIP) under different irradiation spectra: ultraviolet A light (UVA), simulated solar light, and visible light. As expected, all prepared samples demonstrated efficient CIP degradation. For all irradiation sources, increasing synthesis temperature and increasing nitrogen content further improved the degradation efficiencies.

## 1. Introduction

Nowadays, owing to the fast-growing population and the development of industries, where thousands of chemical substances are produced, the risk of an unregulated release of harmful substances into the environment is significant. Among all kinds of pollutants that enter the water bodies every day, organic micropollutants (OMPs) have gained significant attention in the last 20 years due to their impact on the environment [1]. These OMPs, which are mainly pharmaceuticals, personal care products, disinfection by-products, and endocrine disrupters, have been found to be harmful to aquatic life because they are persistent, and some of them bioaccumulate over time, promoting the appearance of antibiotic-resistant bacteria [2,3,4,5]. Additionally, it has been shown that conventional wastewater treatment plants (WWTPs) have problems removing some OMPs, i.e., the conventional WWTPs do not prevent their entry into the environment [6]. Therefore, it is necessary to develop new treatment technologies that can be linked to the existing WWTPs to remove these OMPs efficiently.

Advanced oxidation processes (AOPs) have been considered as an excellent alternative for the removal of OMPs [7,8]. In general, AOPs, such as Fenton, electrochemical oxidation, ozonation, photocatalysis, etc., rely on the generation of reactive oxygen species (ROS) such as hydroxyl radical to oxidize organic pollutants present in the water [9,10,11]. Among different AOPs, TiO_2_ heterogeneous photocatalyst is actively studied because of its outstanding photocatalytic activity, low-cost, excellent chemical stability, and non-toxicity [12,13,14]. However, TiO_2_ is photoactive only under ultraviolet (UV) light irradiation due to its high bandgap (3.2 eV), limiting its application under solar irradiation (4% UV, 48% visible). Moreover, the photogenerated electron-hole pair, that initiates the redox reactions, can quickly recombine, decreasing the photocatalytic activity [15,16,17,18]. Therefore, modification of the TiO_2_ unit cell by means of doping can be a strategy to reduce the recombination rate and/or to redshift its optical response to the visible light range.

Metal doping, such as Pd, Pt, Au, etc., has shown improved photocatalytic activity in the visible range. However, the high metal costs and probability of metal leaching due to low thermal stability, which might represent a health risk, reduce its applicability [12,15,19]. Therefore, non-metal doping, such N, P, S, and C, seem to be more feasible and less expensive to enhance photoactivity under solar spectrum irradiation [20,21,22]. Among non-metal doping, nitrogen is widely studied due to its similarity in atomic radius with oxygen, making more probable the atomic substitution [23]. Nitrogen-doped TiO_2_ has shown an interesting photoactive response under visible light, and this can be attributed to the formation of new energetic levels that reduce the energy bandgap [24]. Additionally, some studies suggest that nitrogen could favor the TiO_2_ anatase phase formation, which is the most photoactive TiO_2_ polymorph [25].

On the other hand, TiO_2_ photoefficiency can be affected by its morphological, optical, and structural properties, which can be to an extent governed by the synthesis method [26,27]. Some of the common conventional synthesis strategies are the sol-gel method, chemical vapor deposition, hydrothermal, etc. The main drawbacks of these methods are the pricy resourcing and long synthesis time [28]. However, the microwave-assisted method first emerged as a non-conventional heating method and was soon recognized as an effective method due to the environmental friendliness, shorter synthesis time, and both homogeneous and selective distribution of heating on greater amounts of samples [29,30].

As a model waste substance, an aqueous solution of ciprofloxacin was used. Ciprofloxacin (CIP) is a fluoroquinolone antibiotic used for the treatment of a wide variety of infections [31,32,33], and it is one of the most often found OMPs in the effluents of wastewaters, surface waters, and drinking waters, with a concentration up to 6.5 mg/L [7]. Due to the lack of its removal by biological systems and due to inaccurate information about its toxicity and bioaccumulation, it was included in the second EU watch list of substances for union-wide monitoring in the field of water policy (Decision (EU) 2018/840 of 5 June 2018) and is still under monitoring in the third EU watch list (Decision (EU) 2020/1161 of 4 August 2020) [34,35]. CIP removal from the wastewaters was attempted by adsorption and biological membrane bioreactors, but the removal efficiencies were substandard [36,37].

Therefore, this study focuses on the synthesis of N/TiO_2_ photocatalyst using the non-conventional microwave-assisted method, evaluating the role of the temperature, reaction time, and nitrogen content on the structural and morphological properties of the synthesized materials, as well as on the photocatalytic activity under different radiation sources.

## 2. Materials and Methods

### 2.1. Materials

Titanium (IV) isopropoxide (TTIP, 97%, Sigma-Aldrich, St. Louis, MO, USA), acetylacetone (≥ 99%, Honeywell, Charlotte, NC, USA), ethanol absolute (p.a., Grammol, Zagreb, Croatia), urea (99.05%, Sigma-Aldrich, St. Louis, MO, USA), Ciprofloxacin (CIP, 98%, Acros Organics, Waltham, MA, USA), and TiO_2_ P-25 (cca. 80% anatase, 20% rutile, Degussa-Hűls Co., Frankfurt, Germany) were used as received without further purification. Deionized water was used throughout the experiments. For the investigation of photocatalytic degradation, a solution of antibiotic ciprofloxacin in deionized water of ultrapure quality (with an electrical conductivity of 0.055 µS/cm at 25 °C, produced with a model GenPure (TKA Co., Niederelbert, Germany) was used.

### 2.2. Microwave-Assisted Synthesis Parameters

For determining the microwave oven parameters in the N/TiO_2_ synthesis, the sol-gel method was combined with the microwave-assisted method, using chemical reagents and ratios as described by Thapa et al. [38]. Briefly, TTIP was mixed with acetylacetone (AcAc). Then, ethanol (EtOH) was added while stirring at room temperature. These reagents were mixed at molar ratio of TTIP:AcAc:EtOH = 0.014:0.039:1.37, and subsequently labeled as solution A. On the other hand, urea (N/Ti molar ratio equal 2) was dissolved in 20 mL deionized water and labeled as solution B. Solutions A and B were added dropwise to 80 mL of deionized water under continuous stirring at room temperature. The final solution was stirred for 1 h at room temperature. Then, the solution was transferred to four Teflon vessels in the microwave (MW) oven (Microwave Reaction System SOLV, Multiwave PRO, Anton-Paar GmbH, Graz, Austria) for thermal treatment at specific temperature and time. Inner pressure and temperature were monitored during the synthesis process. The synthesized material was washed several times with ethanol and water, centrifuged, and dried at 65 °C overnight. The chemical molar ratio of reagents was not evaluated. Instead, two different temperatures (150 °C and 200 °C) and three reaction times (10, 20, and 30 min) were considered as critical parameters for the microwave-assisted method. The obtained materials were labeled as N/TiO_2_ T_t, where T and t are the temperatures and time, respectively, used for the thermal treatment in the MW oven (Table 1).

### 2.3. N/TiO_2_ Microwave-Assisted Synthesis

A similar procedure as described above was used to determine the effect of nitrogen in N/TiO_2_ synthesis. However, the N/Ti molar ratio was varied (0, 1, 2, 4, 12, and 24) to evaluate the effect of nitrogen in the photocatalytic and morphological properties. Therefore, different amounts of urea were dissolved in 20 mL of deionized water and labeled as a solution B, (explained in the previous section). The other reagents were kept at the same molar ratio. Only one temperature and reaction time were used in the MW oven, based on the optimal parameters identified above (200 °C and 10 min). The obtained materials were labeled as N/TiO_2_ x, where x represents the N/Ti molar ratio used (Table 1). Commercial TiO_2_ Degussa P25 was used as a benchmark material for comparison purposes regarding morphological and photocatalytic properties.

### 2.4. Photocatalysts Characterization

For the XRD analysis, a Shimadzu XRD6000 (Shimadzu Corporation, Kyoto, Japan) X-ray diffractometer with CuKα radiation was used. The fixed step scans were collected in the 2θ range 20–60° with steps of 0.02° 2θ and counting time 0.6 s under accelerating voltage of 40 kV and current of 30 mA.

The BET surface area, pore volumes, and pore size distribution were estimated from nitrogen adsorption and desorption isotherm data using an ASAP 2000 apparatus (Micromeritics Corporation, Norcross, GA, USA). Prior to the analysis, the sample was degassed (6.6 Pa) at 150 °C to remove any physically adsorbed species. The pore size distribution of the sample was determined by the Barret–Joyner–Halenda model from the data of the adsorption and desorption branch of the nitrogen isotherms.

FTIR spectra were recorded on a Shimadzu IRSpirit (Shimadzu, Kyoto, Japan) in the range of 400–4000 cm^−1^.

Scanning electron microscopy (SEM) of the materials was performed using a Tescan Vega Easyprobe 3 device (Tescan, Brno, Czech Republic). Energy dispersive spectroscopy (EDS) spectra were recorded with a Bruker XFlash 6|30 detector (Bruker, Billerica, MA, USA) at a working distance of 10 mm under accelerating voltage of 20 keV on uncoated samples glued to the stage using carbon two-side adhesive tape. EDS mapping was performed for Ti, O, and N.

Raman measurements were performed by confocal micro-Raman spectroscopy using a T64000 (Horiba Jobin Yvone, Kyoto, Japan) equipped with a solid-state laser with a wavelength of 532.5 nm and a 50× magnification large working distance objective in the range of 90–2000 cm^−1^. Laser power at the sample was optimized to avoid heating and possible phase transition of TiO_2_ during measurement.

DRS measurements were performed on an Ocean Insight QE Pro High-Performance Spectrometer (Ocean Insight, Orlando, FL, USA) equipped with an integrating sphere for reflectance and a DH 2000 deuterium-halogen source in the range 250–950 nm with a resolution of 1 nm and integration time of 10 s.

The chemical composition and bonding were characterized by XPS in a SPECS XPS spectrometer equipped with a Phoibos MCD 100 electron analyzer (SPECS, Berlin, Germany) and a monochromatized source of Al Kα X-rays of 1486.74 eV. The typical pressure in the UHV chamber during analysis was in the 10^−7^ Pa range. For the electron pass energy of the hemispherical electron energy analyzer of 10 eV was used in the present study, and the overall energy resolution was around 0.8 eV. All spectra were calibrated by the position of C 1s peak fixed at the binding energy of 284.5 eV. The XPS spectra were deconvoluted into several sets of mixed Gaussian–Lorentzian functions with Shirley background subtraction.

### 2.5. Photocatalytic Activity Evaluation

The photocatalytic activity evaluation was performed through the degradation of ciprofloxacin, using three different radiation sources: (I) ultraviolet A (UVA) lamp, model UVAHAND LED (Dr. Hönle AG, UV-Technologie, Gilching, Germany) (peak on 365 nm, 70 W), (II) solar light simulator (SLS), model SOL500 (Dr. Hönle AG, UV-Technologie, Gilching, Germany) (430 W), and (III) warm visible light lamp, model OSRAM Endura Flood 50 W WT, (Ledvance GmbH, Osram, Munich, Germany) (peaks on 450 nm and 600 nm, 50 W). In each experiment, 25 mg of the photocatalyst was dispersed in 100 mL of CIP solution (10 mg/L) and irradiated from above with lamps 20 cm away from the reactor. First, the suspension was stirred for 30 min in the dark to ensure adsorption-desorption equilibrium, which was determined previously by adsorption test. Then, the lamp was turned on and irradiated the suspension for 2 h. Samples were taken from the reactor at intervals of 0, 5, 10, 20, 30, 45, 60, 90, and 120 min, filtered using a 0.45 µm mixed cellulose ester membrane filter, and analyzed with a UV-Vis spectrophotometer (HEWLETT PACKARD, Model HP 8430, Palo Alto, CA, USA) at 273 nm (maximum absorption peak of CIP). During the whole experiment, the temperature was kept at 25 °C by a thermostatic bath. All radiation spectra of the used lamps and UV-VIS absorbance of the 10 mg/L solutions of the CIP in water are given in Appendix A.

## 3. Results & Discussion

### 3.1. Characterization of N/TiO_2_ for Optimization of the Reaction Time and Temperature

To optimize the sample preparation in terms of the reaction temperature and time, the N/TiO_2_ samples with constant N ratio were subjected to broad structural, microstructural, and spectroscopic characterization. For prepared N/TiO_2_ materials, the nitrogen adsorption/desorption isotherms were collected and described by the Brunauer–Emmett–Teller (BET) and Barrett–Joyner–Halenda (BJH) models to derive specific surface area, pore volume, and pore size distribution values (Table 2). It can be observed that N/TiO_2_ materials synthesized at different reaction times in the MW have a similar specific surface area, pore-volume, and average pore diameter, indicating that the reaction time on the MW oven does not have a significant effect on the porosity of the material. However, increasing the reaction temperature for a reaction time of 10 min (N/TiO_2_ 150_10 and N/TiO_2_ 200_10) results in an increase in specific surface area, pore volume, and average pore diameter. The FTIR spectra of the N/TiO_2_ materials synthesized at different temperatures and reaction times are presented in Figure 1a. All materials show a strong wide band between 400 and 800 cm^−1^, which corresponds to the stretching vibrations of Ti–O–Ti bonds [39], whereas the broad band between 3650 and 2850 cm^−1^, and the peak at 1632 cm^−1^, are attributed to stretching vibrations of O–H group [40]. However, N/TiO_2_ 150_10 has additional peaks at 1590, 1525, 1443, and 1372 cm^−1^, which are ascribed to the stretching vibrations of C=O, N–H, and C–N groups, respectively, associated with the residual urea in the material [41]. While indicative for composition, FTIR hardly facilitates the selection of optimal reaction temperature and time. Figure 1b shows the X-ray diffraction patterns of N/TiO_2_ materials synthesized at different temperatures and reaction times. All materials show diffraction peaks of TiO_2_ anatase, assigned to the ICDD PDF#21–1272 [42]. It can be noticed that a higher treatment temperature is leading to narrower peaks, indicating larger crystallite size. Similar results were reported by Kadam et al. [43]. However, in this study, the anatase crystal phase is formed even at temperatures as low as 150 °C. Contrary to reaction temperature, the reaction time has a negligible effect on the crystal phase and the crystallite size (Figure 1b).

### 3.2. Photocatalysis of N/TiO_2_ for Optimization of the Reaction Time and Temperature

A photocatalytic experiment using the N/TiO_2_ samples with constant N ratio was utilized for optimization of the sample preparation in terms of the reaction temperature and time, similarly to the characterization. The photocatalytic activity of the N/TiO_2_ materials synthesized at different temperatures and reaction times was evaluated for the degradation of CIP aqueous solution (10 mg/L) under the UVA light and visible light, as shown in Figure 2. The photocatalytic experiments under the UV light (Figure 2a) show that N/TiO_2_ materials synthesized at 200 °C degrade 90% of CIP in just 20 min of irradiation, while the material synthesized at 150 °C achieves less than 80% of removal at the same period. Additionally, it is noticed that materials synthesized at different reaction times (10, 20, and 30 min of MW oven treatment) have similar photoactivity, comparable to commercial Degussa P25 TiO_2_, achieving more than 95% degradation of CIP in only 45 min under the UV radiation with lamp (I). After 45 min, no noticeable CIP degradation changes are detected for materials synthesized at 200 °C, while the material synthesized at 150 °C requires more than 90 min to reach similar removal efficiency (Figure 2c). Under the visible light, all N/TiO_2_ materials show some photoactivity under lamp III, while the photocatalyst Degussa P25 TiO_2_ shows no activity at all (Figure 2b). The reason for photoactivity under the visible light is most likely the presence of nitrogen, which might cause the narrowing of the bandgap of TiO_2_, shifting its photo-responses to the visible spectrum [23,39]. In the CIP removal under both radiation sources, all synthesized materials show a synergistic effect of adsorption and photocatalysis, as shown in Figure 2c. The N/TiO_2_ synthesized at 150 °C displays the highest adsorption capacity, which could be due to the affinity between amino or carboxylic group in the ciprofloxacin structure and the urea remaining in the material synthesized at 150 °C, as confirmed by the FTIR analysis. Despite the fact that the material synthesized at 150 °C displays a higher adsorption capacity, it has lower photoactivity than the materials synthesized at 200 °C. The materials synthesized at 200 °C and at different reaction times show similar photoactivity under both radiation sources (UVA and warm visible lights).

Based on the characterization and photocatalytic results, the photoactivity is mainly affected by the size of the crystallite domains, which is governed by the synthesis temperature. The synthesis time marginally affected the morphological and photocatalytic properties. Therefore, 200 °C for 10 min is elucidated as the optimal MW oven treatment for further N/TiO_2_ materials synthesis, where the amount of urea was optimized.

### 3.3. Bulk Characterization of N/TiO_2_ for Optimization of the N Doping Amount

Following the elucidated optimal reaction time and temperature, we subjected the samples to broad structural, microstructural, and spectroscopic characterization in order to optimize the preparing procedure in terms of the N content in the N/TiO_2_ samples. The specific surface area, pore-volume, and average pore diameter of Degussa P-25 and N/TiO_2_ materials synthesized at different N/Ti molar ratios are presented in Table 3. It can be noticed that the specific surface areas of N/TiO_2_ materials are reduced while increasing the N/Ti ratio. The sample without urea (N/TiO_2_ 0) presents the highest specific surface area. A similar reduction of the specific surface area with the increment of nitrogen ratio was reported by Ma et al. [44]. This reduction could be attributed to interstitial doping, in which nitrogen atoms could be located in the interstitial voids instead of crystal lattice sites [21]. Moreover, the specific surface areas of N/TiO_2_ materials are significantly higher than the photocatalyst Degussa P-25 (up to four times higher). Interestingly, it can be observed that the pore volume and the average pore diameter are not significantly affected by the N/Ti ratio. For the case of N content variations, obviously, it is not as straightforward to determine the optimal content.

In Figure 3a, the FTIR spectra of the N/TiO_2_ materials and Degussa P25 are presented. A strong and wide band between 400 and 800 cm^−1^ is observed for all materials, which can be associated with the stretching vibrations of Ti–O–Ti bonds. Additionally, the wide band between 3650 and 2850 cm^−1^ and the peak at 1632 cm^−1^ are correlated to stretching vibrations of the O–H group. Furthermore, in the N/TiO_2_ materials containing a high N/Ti molar ratio (12 and 24), a small band around 1457 cm^−1^ appears, which is attributed to the vibrations of the formed Ti-N bond [45]. Interestingly, all synthesized materials have a higher intensity of bands related to the O–H group than the P25 sample, which could be beneficial for the production of hydroxyl radicals during photocatalytic reactions [46,47]. The optimal N content can be hardly elucidated from the FTIR scans. The Raman spectra of the N/TiO_2_ materials and commercial photocatalyst Degussa P25 are shown in Figure 3b. In general, for all materials, five peaks can be identified which are associated with the A_1g_ (515 cm^−1^), B_1g_ (396 cm^−1^) and E_g_ (143, 396, and 637 cm^−1^) Raman modes of TiO_2_ anatase phase [22,48], while, for the Degussa P25, an additional small peak at 444 cm^−1^, which is characteristic for the TiO_2_ rutile phase, is observed [49]. In all N/TiO_2_ synthesized materials (with and without urea), a slight shifting of the Raman peaks of the B_1g_ and E_g_ to higher wavenumbers is detected, and it could be attributed either to changes of surface oxygen vacancies or incorporation of nitrogen into the TiO_2_ structure [24]. The Raman spectra indicate that nitrogen does not affect the crystallization of the TiO_2_. Figure 3c shows the X-ray diffraction patterns of N/TiO_2_ materials and commercial photocatalyst Degussa P25. All synthesized N/TiO_2_ materials display only diffraction peaks assigned to anatase (ICDD PDF#21–1272) [42], with crystallite size between 13 and 15 nm (determined by the Scherrer method); while the TiO_2_ Degussa P25 exhibits diffraction peaks are assigned to anatase (ICDD PDF#21–1272) and rutile (ICDD PDF#21–1276) [50], with average crystallite size of about 30 nm. In the materials containing urea, the anatase diffraction peak (101) is slightly shifted to lower angle (2θ = 25.18°) compared to material without urea (2θ = 25.41°), as shown in Figure 3d, and it could be associated with nitrogen doping [30]. Even though N content variations were used, there are no observable changes in the phase composition nor the crystallite size of the N/TiO_2_ materials. Apart from the slight shifting of the diffraction peak (101), no change is detected between materials with and without urea, proving the negligible structural effect of the nitrogen doping.

The SEM analysis was performed on the materials synthesized at 0 and 12 N/Ti molar ratios (Figure 4). Agglomerated particles having spherical shape are observed for both materials. It is observed that nitrogen addition does not have an effect on the morphology, shape, or size of the TiO_2_ particles. In Figure 5, the fit of EDS analysis for materials synthesized at 0 and 12 N/Ti molar ratios, respectively, are presented. The Ti and O signals, which are typical in TiO_2_, are detected in both materials. Nitrogen traces are present even in the N/TiO_2_ 0, which can be due to the nitrogen from the atmosphere because materials were stored at atmospheric conditions. In addition, it is well known that for values below 5%, quantitative analysis of the EDS spectra is less reliable. However, relatively more nitrogen is evident in the material N/TiO_2_ 12. High-resolution EDS spectra were acquired precisely and fitted to allow more reliable quantitative results compared to conventional automated EDS analysis. Moreover, it is observed that EDS mapping of N for the material N/TiO_2_ 12 points out a stronger and more homogeneous presence of N throughout the material, which is in line with the applied synthesis conditions.

### 3.4. Surface Specific Characterization of N/TiO_2_ for Optimization of the N Doping

In Figure 6, the high-resolution spectra of Ti, O, and N obtained by XPS measurements for N/TiO_2_ 0 and N/TiO_2_ 12 are shown, in which the chemical composition and oxidation states were determined. In Figure 6a,c, the O 1s spectra of materials N/TiO_2_ 0 and N/TiO_2_ 12, respectively, are shown. Three components with different binding energy values are observed in both materials, where the main peak at 530.1 eV is assigned to oxygen bonded to titanium (O-Ti). The peak at 532.8 eV is attributed to adsorbed oxygen, probably for O-H bonds of chemisorbed water [44]. Additionally, it is noticed that the peak related to chemisorbed water is slightly more intense in the sample N/TiO_2_ 12, which favors the photocatalytic reactions due to the higher probability of generating of hydroxyl radicals. On the other hand, the third peak at 531.6 eV could be related to oxygen bonded to carbon, which could present impurities from the synthesis (urea or acetylacetone). Figure 6b,d show the Ti 2p spectra of materials N/TiO_2_ 0 and N/TiO_2_ 12. In both materials, two peaks at 458.8 eV and 464.5 eV, corresponding to Ti 2p^1/2^ and Ti 2p^3/2^, are found. These two signals are evidence of the formation of the Ti^4+^, related to the anatase phase of TiO_2_ [43,51,52]. Figure 6e represents N 1s spectrum recorded for N/TiO_2_ 12. The small peak with binding energy of 400.15 eV is attributed to nitrogen bonded to oxygen (N-O) [45,53]. In general, it is assumed that the N 1s peak above 400 eV is related to interstitial doping [24]. The introduction of nitrogen in the TiO_2_ lattice produces slight changes in the electronic densities of Ti and O. For the O 1s spectra, the main peak is shifted to lower binding energy from 530.1 to 529.95 eV; while for the Ti 2p spectra, the energy for the Ti 2p^1/2^ is shifted to lower binding energy from 458.8 to 458.7 eV. The binding energy shifting is higher for the O 1s spectra due to N-O bonding. Usually, N-O bonding in the interstitial sites is achieved under wet chemical processes, which is the applied method used in the present study [54]. Although there are several techniques that can help to determine the presence of a doping element such as nitrogen, the XPS analysis is the most suitable technique for confirming the success of the doping process as well as the type of doping (substitutional or interstitial) [21].

The DRS results and the Tauc plots are shown in Figure 7. From the DRS analysis (Figure 7a), it is observed that there are no significant differences in the energy absorption for the N/TiO_2_ materials and Degussa P25. All materials only absorb photons at wavelengths shorter than 400 nm. Based on the DRS, the Tauc plots can be obtained to determine the energy bandgap of all materials (Figure 7b). It is noticed that all N/TiO_2_ materials display a slight shift to the lower energy absorption in comparison to the Degussa P25. Additionally, it is observed that the increment in the N/Ti molar ratio increases the energy bandgap. Similar results were reported by Suwannaruang et al. [55]. The lowest energy bandgap is obtained for the material N/TiO_2_ 0. The increment in the energy bandgap could be explained by the fact that the nitrogen is bonded to oxygen and it is introduced interstitially, as confirmed by XPS analysis, where nitrogen acts as an impurity, creating a new energetic level that contributes to the absorption in the visible light range without modifying the valence band (VB) or conduction band (CB) of TiO_2_ [54,56]. Kuo et al. confirmed by density functional theory calculations that band gap narrowing is achieved by substitutional nitrogen doping, while interstitial nitrogen doping produces localized impurities that do not reduce the energy band gap. However, both types of doping contributes to the absorption of photons in the visible range [56]. Previous studies have shown that a lower energy bandgap is not the only condition to achieve a higher photoactivity in the visible region. Other factors, such as oxygen vacancies and new energetic levels in the TiO_2_ lattice, also play an important role in photocatalytic activity [50,54,55,57].

### 3.5. Photocatalytic Performance Experiment for N/TiO_2_ Samples for Optimization of N Content

The photoactivity of the N/TiO_2_ materials with the different N/Ti molar ratios, synthesized at 200 °C for 10 min in the MW oven, were evaluated again through the degradation of CIP aqueous solution (10 mg/L) under the UVA light (lamp I), solar light simulator (lamp II) and visible light (lamp III), as shown in Figure 8. In the Appendix A, the additional photocatalytic degradation for materials with 1 and 2 N/Ti molar ratios (Appendix A) are shown. The photocatalytic data are described using the pseudo first-order and pseudo second-order models. The pseudo first-order model described a process where the degradation rate is affected mainly by the changes in pollutant concentration, while the pseudo second-order model described a process where the degradation rate is affected, besides the pollutant concentration, by other factors, such as light intensity or intermediates formation [58]. The kinetic constant for the pseudo first-order is obtained by the slope of the plot of -Ln (*C*/*C*o) versus the irradiation time. For the pseudo second-order, the kinetic constant is obtained by the slope of the plot of (1/*C* -1/*C*o) versus the irradiation time. Analysis of the goodness of fit parameters for each model should point to one offering the better description of the process and thus better pointing out to reaction mechanism. Table 4 and Table 5 show the pseudo first-order (*k_1_*, min^−1^) and pseudo second-order kinetic constants (*k_2_*, L mg^−1^ min^−1^) and removal efficiencies of N/TiO_2_ materials. The pseudo first-order model shows that under the UV and solar light, the correlation coefficient (*R*^2^) in most of the cases has value below 0.90, indicating that the model does not describe well the experimental degradation process. On the contrary, the pseudo second order in all cases presents a correlation coefficient (*R*^2^) around 0.99, indicating that the model describes more precisely the degradation process under the UV and solar light. In the case of photocatalytic degradation under the visible light, although both models have a correlation coefficient (*R*^2^) above 0.90, the pseudo second-order fits better to the experimental results. From the photocatalytic tests, it is noticed that materials with a higher N/Ti molar ratio (12 and 24) and the photocatalyst Degussa P-25 achieve up to 90% CIP removal in just 20 min of UVA (lamp I) irradiation, while the materials with lower N/Ti molar ratio and N/TiO_2_ 0 achieve less than 90% of removal in the same period. After 60 min of UVA irradiation, no significant changes in the CIP concentration are detected for all N/TiO_2_ materials and Degussa P-25 (Figure 8a). Moreover, it is noticed that by increasing N/Ti molar ratio, the degradation rate is increased, as displayed in Table 5. Under the SLS radiation (lamp II), similar behavior as under the UVA radiation is observed, where just 20 min of irradiation is required for all materials to achieve 90% of CIP removal. It is observed that an increment N/Ti molar ratio increases the degradation rate under the SLS radiation. However, the kinetic constants for all N/TiO_2_ materials are higher under the SLS than under the UVA radiation. On the contrary, the commercial photocatalyst Degussa P-25 shows a lower kinetic constant under the SLS than under the UVA radiation (Table 5). The improvement in the kinetic constants under the SLS could be attributed to new energetic levels in the TiO_2_ lattice due to the nitrogen doping, as confirmed by XPS analysis. The energetic levels due to nitrogen could shift the light absorption to the visible range and promote the charge separation, enhancing the photocatalytic activity under the solar spectrum [21,48]. On the other hand, the maximum degradation rate is achieved for the N/TiO_2_ 12 under the UVA (lamp I) and solar radiation (lamp II). Having a N/Ti molar ratio higher than 12, the kinetic constant decreases due to the fact that nitrogen excess could act as a recombination center that reduces photocatalytic activity [44]. The CIP degradation under the warm visible light (lamp III) shows that all N/TiO_2_ materials have photocatalytic activity, while the commercial TiO_2_ does not show photoactivity under this irradiation source (Figure 8c). Again, it is observed that photoactivity rate increases when N/Ti molar ratio also increases. However, after 120 min of irradiation, any N/TiO_2_ material achieves a complete CIP removal under this radiation source. This could be explained by the fact that the photons emitted by the lamp III are mainly from the wavelength around 600 nm, and just a small portion of photons are emitted at 450 nm (Appendix A), making the kinetic constants 100 times lower than under the SLS. Therefore, a longer irradiation period could be required to achieve complete CIP removal.

### 3.6. Determining the Relation Adsorption/Photocatalytic Removal of CIP for Different Irradiation Sources

Under the three different radiation sources, a synergistic effect of adsorption and the photocatalytic process takes place for CIP removal, as shown in Figure 8 and Figure 9. Regardless of the N/Ti molar ratio, all N/TiO_2_ materials display similar adsorption capacity, where the adsorption process represents around 30% of CIP removal. This similar adsorption capacity is related to the high specific surface area of all N/TiO_2_ materials and due to the affinity of the amine and carboxylic groups of CIP with the hydroxyl groups in the surface of the materials, favoring the adsorption of the pollutant on the photocatalyst surface. Comparing and evaluating the results from different models (pseudo first-order and pseudo second-order) leads to the conclusion that the photocatalytic degradation of CIP by the N/TiO_2_ materials follows the second order kinetic model, similar to the results reported by Gabelica et al. [28]. The pseudo first-order and pseudo second-order kinetic plots of CIP photocatalytic degradation by Degussa P25 TiO_2_ and N/TiO_2_ materials are given in Appendix A, respectively.

## 4. Conclusions

Here, we demonstrate the successful utilization of the microwave-assisted synthesis for the purpose of preparing nitrogen doped titania nanoparticles. This non-conventional approach has proven advantages on the basis of a documented reduction of the environmental impact due to the reduction of necessary energy, solvent, and stabilizers compared to majority of other wet-chemistry synthesis techniques. We employed a strategy for preparing, characterizing, and optimizing the preparation in two cycles, for the purpose of selecting the best reaction temperature, time, and N content. Samples are crystallized as anatase without subsequent thermal treatment, and the reaction temperature was identified as the main quality parameter. On the other hand, the change of the composition, i.e., precursor ratio, hardly influenced the morphology of the samples. However, N content was important for photocatalytic performance.

The photocatalytic activity of N/TiO_2_ nanoparticle containing aqueous suspensions was evaluated for the degradation of antibiotic ciprofloxacin (CIP) under different irradiation spectra: UVA, simulated solar light, and visible light. For all experiment conditions and samples investigated, the degradation performance was efficient and relatively aligned. Increasing the synthesis temperature and nitrogen content seems to be beneficial for increasing the degradation efficiency up to a point of saturation, after which nitrogen can act as a recombination center. Finally, the overall best material proved to be TiO_2_ derived at 200 °C for 10 min, having an intermediate N/Ti molar ratio of up to 12 using the synergy of three irradiation sources.

## Figures and Tables

**Figure 1 nanomaterials-12-01473-f001:**
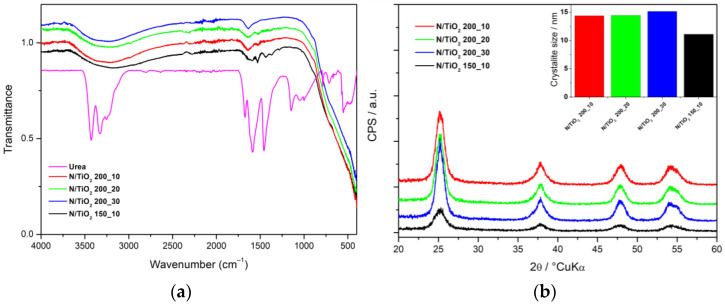
(**a**) FTIR spectra of N/TiO_2_ materials, and (**b**) X-ray diffraction patterns, crystallite size.

**Figure 2 nanomaterials-12-01473-f002:**
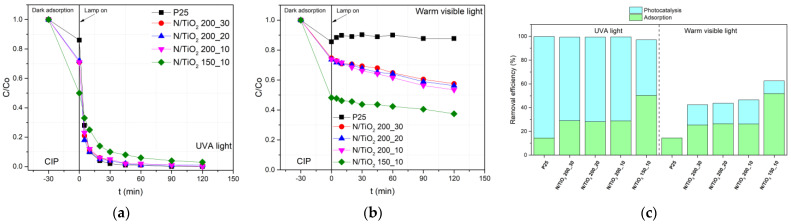
Photocatalytic degradation of ciprofloxacin (CIP) by Degussa P25 TiO_2_ and N/TiO_2_ materials under the (**a**) ultraviolet A (UVA) light (lamp I) and (**b**) warm visible light (lamp III). (**c**) CIP Removal efficiencies by Degussa P25 TiO_2_ and N/TiO_2_ materials under the UVA and warm visible lights.

**Figure 3 nanomaterials-12-01473-f003:**
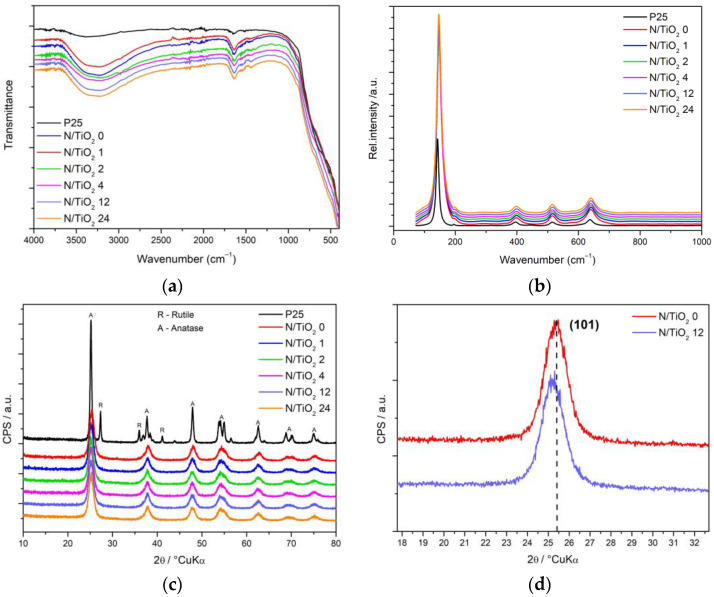
(**a**) FTIR spectra of N/TiO_2_ materials and Degussa P25; (**b**) Raman spectra; (**c**) X-ray diffraction patterns, and; (**d**) X-ray diffraction peak (101) of the N/TiO_2_ materials synthesized at 0 and 12 N/Ti molar ratios.

**Figure 4 nanomaterials-12-01473-f004:**
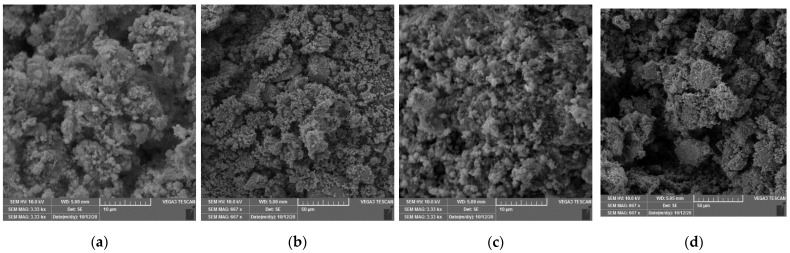
(**a**,**b**) SEM images of N/TiO_2_ 0. (**c**,**d**) SEM images of N/TiO_2_ 12.

**Figure 5 nanomaterials-12-01473-f005:**
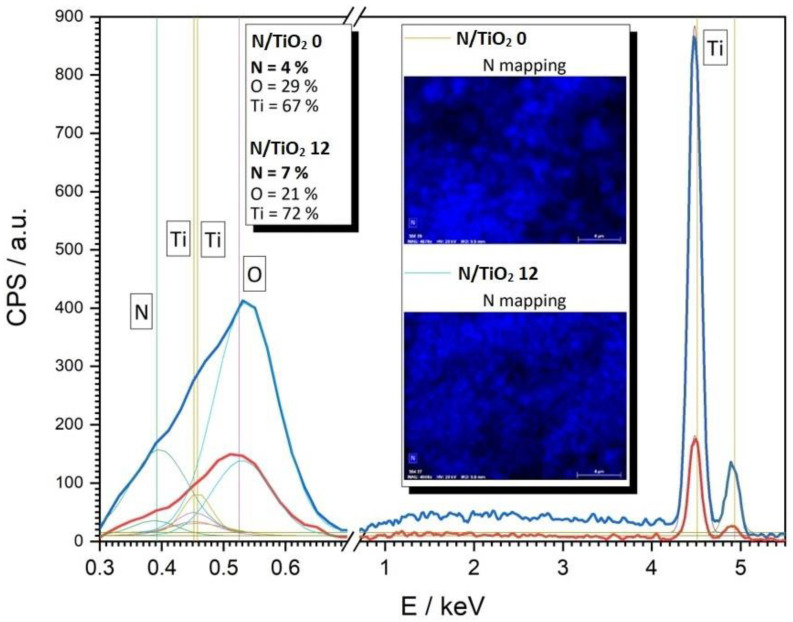
The fit of the EDS spectra for the N/TiO_2_ 0 (4 atomic % of N) and N/TiO_2_ 12 (7 atomic % of N). Inset: EDS mapping for the TiO_2_ materials synthesized at 0 and 12 N/Ti molar ratios.

**Figure 6 nanomaterials-12-01473-f006:**
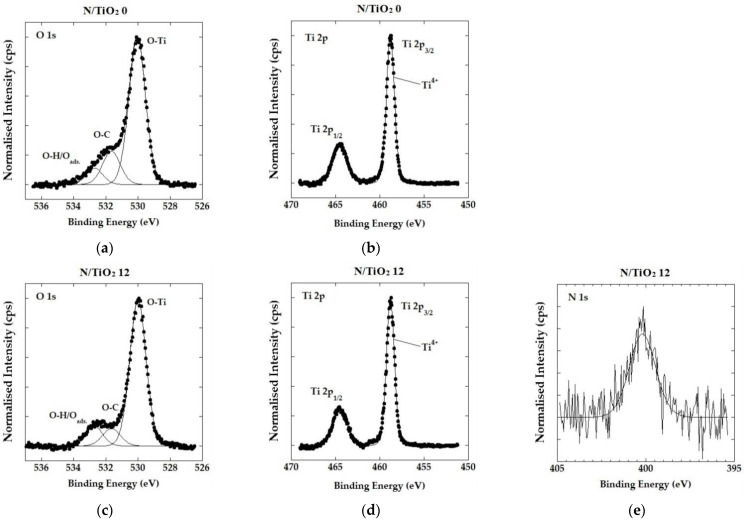
High-resolution XPS spectra: (**a**) O 1s spectrum, and; (**b**) Ti 2p spectrum, N/TiO_2_ 0; (**c**) O 1s spectrum; (**d**) Ti 2p spectrum, and; (**e**) N 1s spectrum N/TiO_2_ 12.

**Figure 7 nanomaterials-12-01473-f007:**
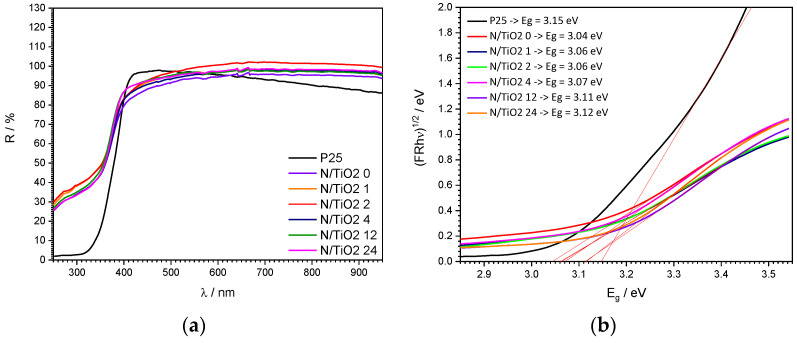
(**a**) DRS spectra and (**b**) Tauc plot for energy bandgap determination of N/TiO_2_ materials and Degussa P25.

**Figure 8 nanomaterials-12-01473-f008:**
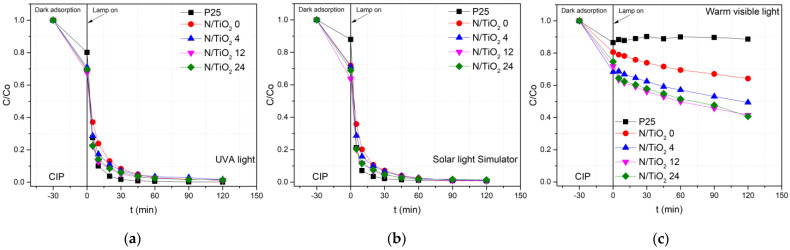
Photocatalytic degradation of ciprofloxacin by Degussa P25 TiO_2_ and N/TiO_2_ materials under (**a**) UVA light (lamp I), (**b**) Solar light simulator lamp (II), and (**c**) warm visible light (lamp III).

**Figure 9 nanomaterials-12-01473-f009:**
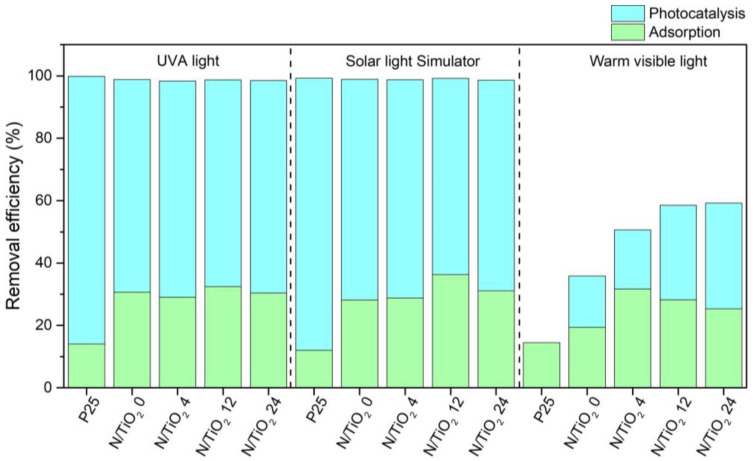
CIP removal efficiencies by Degussa P25 TiO_2_ and N/TiO_2_ materials under UVA light (lamp I), Solar light simulator lamp (II), and warm visible light (lamp III).

**Table 1 nanomaterials-12-01473-t001:** Material labeling and evaluated parameters.

Material Labeling	Parameter	Section
N/TiO_2_ 150_10	Temperature (from 150 to 200 °C) and time (from 10 to 30 min)	Microwave-assisted synthesis parameters (Section 2.2)
N/TiO_2_ 200_10		
N/TiO_2_ 200_20		
N/TiO_2_ 200_30		
N/TiO_2_ 0	Nitrogen content (from 0 to 24 N/Ti molar ratio)	N/TiO_2_ microwave-assisted synthesis parameters (Section 2.3)
N/TiO_2_ 1		
N/TiO_2_ 2		
N/TiO_2_ 4		
N/TiO_2_ 12		
N/TiO_2_ 24		

**Table 2 nanomaterials-12-01473-t002:** Specific surface area, pore-volume, and crystallite size of N/TiO_2_ materials.

Material	*S*_BET_, m^2^ g^−1^	*V*_pore_, cm^3^ g^−1^	Average Pore Diameter, nm
N/TiO_2_ 150_10	172.4	0.260	5.46
N/TiO_2_ 200_10	185.3	0.326	6.68
N/TiO_2_ 200_20	187.8	0.342	6.91
N/TiO_2_ 200_30	179.2	0.330	7.02

**Table 3 nanomaterials-12-01473-t003:** Specific surface area, pore-volume, and pore size of N/TiO_2_ materials and Degussa P25.

	*S*_BET_, m^2^ g^−1^	*V*_pore_, cm^3^ g^−1^	Average PoreDiameter, nm
N/TiO_2_ 0	215.04	0.424	7.25
N/TiO_2_ 1	144.40	0.273	7.26
N/TiO_2_ 2	185.29	0.326	6.68
N/TiO_2_ 4	158.09	0.295	7.19
N/TiO_2_ 12	139.17	0.297	8.00
Degussa P25	48.14	0.196	13.69

**Table 4 nanomaterials-12-01473-t004:** Parameters obtained from the ciprofloxacin (CIP) degradation under UVA (lamp I), Solar light simulator (lamp II), and warm visible light (lamp III) fitted to a pseudo-first-order kinetic model for N/TiO_2_ materials.

	*k_1_*, (min^−1^)			*R^2^*		
Lamp	I	II	III	I	II	III
N/TiO_2_ 0	0.0380	0.0432	0.0021	0.9001	0.9023	0.9796
N/TiO_2_ 4	0.0406	0.0443	0.0028	0.8893	0.8928	0.9957
N/TiO_2_ 12	0.0432	0.0477	0.0053	0.8635	0.8843	0.9432
N/TiO_2_ 24	0.0425	0.0464	0.0055	0.8621	0.8673	0.9392
Degussa P25	0.0588	0.0563	-	0.8562	0.8187	-

**Table 5 nanomaterials-12-01473-t005:** Parameters obtained from the CIP degradation under UVA (lamp I), Solar light simulator (lamp II), and warm visible light (lamp III) fitted to a pseudo-second-order kinetic model, and removal efficiency of N/TiO_2_ materials.

	*k_2_*, (L mg^−1^ min^−1^)			*R^2^*			Removal Efficiency (ɳ)		
Lamp	I	II	III	I	II	III	I	II	III
N/TiO_2_ 0	0.3905	0.5750	0.0028	0.9959	0.9909	0.9861	98.07%	98.68%	35.83%
N/TiO_2_ 4	0.4798	0.6499	0.0044	0.9946	0.9886	0.9978	98.46%	98.78%	50.64%
N/TiO_2_ 12	0.5881	0.9768	0.0088	0.9975	0.9869	0.9687	98.66%	99.21%	58.51%
N/TiO_2_ 24	0.5376	0.7852	0.0091	0.9872	0.9974	0.9655	98.40%	99.02%	59.26%
Degussa P25	1.7868	1.2320	-	0.9907	0.9948	-	99.61%	99.35%	12.32% *****

***** By adsorption process.

## Data Availability

The data presented in this study are available upon reasonable request from the corresponding author.

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
