# Peer review of "Microwave-Assisted Synthesis of N/TiO2 Nanoparticles for Photocatalysis under Different Irradiation Spectra"

_nanomaterials, 2022, doi:10.3390/nano12091473_

Round 1

Reviewer 1 Report

Dear Editor, in this manuscript, Tobon et al., highlighted “Microwave-assisted synthesis of N/TiO2 nanoparticles for photocatalysis under different irradiation spectra”. I think the paper has been formulated in a good manner. Samples are adequately characterized, and the removal of antibiotic has been discussed in detail. However, the manuscript still has many problems and is not suitable for publication at this stage. I, therefore, suggest major review for it before its acceptance. My questions are given below.

1) My first concern is about the English language of the manuscript which is poor. There is correlation between the abstract, experimental procedure and result and discussion. Normally, the abstract and experimental procedure are written in past tense while result and discussion in present tense. Please double check it with a native English speaker. For example, “From the DRS analysis (Figure 7a) it was observed that there are no significant differences…. Similarly, many mistakes are seen in the whole manuscript.

2) The amount of N2 introduced is very high and normally the dopant concentration is low. Can the author explain that why this huge amount oof N2 was selected? Why was the band gap not altered?

3) What do the electronic densities of Ti and N change during doping as not evident from the XPS data of the samples? The author must explain in detail.

4) It is accepted that orbital energy of O is lower than N, oxygen being more electronegative than N, therefore, the excitation of electrons from O requires more energy than electronic transition from N, and therefore Band gap must decrease as the concentration of N is increased or N is introduced in TiO2. However, the situation is different here. Please explain it in detail.

5) Why does the author remove antibiotic from water? What are the applications of antibiotics to produce drug-resistance bacteria? For more detail I suggest the citation of papers like a) Z. Zhang, A. Zada, N. Cui, N. Liu, M. Liu, Y. Yang, D. Jiang, J. Jiang, S. Liu, Synthesis of Ag loaded ZnO/BiOCl with high photocatalytic performance for the removal of antibiotic pollutants, Crystals 11 (2021) 981. b) N. Cui, A. Zada, J. Song, Y. Yang, M. Liu, Y. Wang, Y. Wu, K. Qi, R. Selvaraj, S. Liu, G. Jin, Plasmon-induced ZnO-Ag/AgCl photocatalyst for degradation of tetracycline hydrochloride, Desalination Water Treat. 245 (2022) 247-254.

6) What is the effect of N doping on the charge separation of TiO2 as included in the shortcomings of TiO2 by the author? The author has not claimed that any where in the manuscript. What experiments were performed to measure the charge separation?

7) Cu and Fe are not costly metals therefore please make necessary correction.

8) What about the stability of the doped samples? N2 definitely decreases stability of TiO2.

9) Some very important citations are missing.

a) Zada, M. Khan, Z. Hussain, M. I. A. Shah, M. Ateeq, M. Ullah, N. Ali, S. Shaheen, H. Yasmeen, S. N. A. Shah, A. Dang, Extended visible light driven photocatalytic hydrogen generation by electron induction from g-C3N4 nanosheets to ZnO through the proper heterojunction, Z. Phys. Chem. 236 (2022) 53-66.

Author Response

Thank you for the review. We have done our best to correct the article and fulfill all the requirements and questions this time. We are resubmitting the corrected manuscript as suggested by the reviewers. Please see the attachment, in which point-by-point response is provided. 

Camilo Sanchez Tobon

Reviewer 2 Report

My comments are in attached file.

Author Response

(The authors gave the same response as above.)

Round 2

Reviewer 1 Report

Since the author responded positively to improve the quality of the paper, therefore, I suggest the acceptance of this paper in your Journal.